# Household costs associated with zoonotic *Plasmodium knowlesi, P. falciparum, P. vivax* and *P. malariae* infections in Sabah, Malaysia

**Patrick Abraham**[1], **Campbell McMullin**[1], **Timothy William**[2], **Giri S. Rajahram**[2,3,4], **Jenarun Jelip**[5], **Roddy Teo**[6], **Chris Drakeley**[7], **Abdul Marsudi Manah**[8], **Nicholas M. Anstey**[9], **Matthew J. Grigg**[9☯], **Angela Devine**[1,9☯]*

**1** Melbourne Health Economics, Centre for Health Policy, Melbourne School of Population and Global Health, The University of Melbourne, Melbourne, Victoria, Australia, **2** Infectious Disease Society-Menzies School of Health Research, Kota Kinabalu, Sabah, Malaysia, **3** Queen Elizabeth Hospital II, Ministry of Health Malaysia, Kota Kinabalu, Sabah, Malaysia, **4** School of Medicine and Health Sciences, Monash University Malaysia, Kuala Lumpur, Malaysia, **5** Vector Borne Disease Sector, Ministry of Health Malaysia, Putrajaya, Malaysia, **6** Public Health Research Section, Sabah State Department of Health, Ministry of Health Malaysia, Kota Kinabalu, Sabah, Malaysia, **7** London School of Hygiene & Tropical Medicine, London, United Kingdom, **8** Vector Borne Unit, Sabah State Department of Health, Ministry of Health Malaysia, Kota Kinabalu, Sabah, Malaysia, **9** Global and Tropical Health Division, Menzies School of Health Research, Charles Darwin University, Darwin, Northern Territory, Australia

☯ These authors contributed equally to this work.
* angela.devine@menzies.edu.au

## Abstract

### Background

Malaysia has free universal access to malaria care; however, out-of-pocket costs are unknown. This study estimated and compared household costs of illness during a unique time when four species of malaria were present, due to the emergence of zoonotic *Plasmodium knowlesi* during the elimination phase of non-zoonotic species in Sabah, Malaysia.

### Methodology/principal findings

Household costs were estimated from patient-level surveys collected from four hospitals between 2013 and 2016. Direct costs including medical and associated travel costs, and indirect costs due to lost productivity were included. One hundred and fifty-two malaria cases were enrolled: *P. knowlesi* (n=108), *P. vivax* (n=22), *P. falciparum* (n=16), and *P. malariae* (n=6). Costs were inflated to 2023 Malaysian Ringgits and reported in United States dollars (US$). Across all cases, the mean total costs were US$131 (SD=102), with productivity losses accounting for 58% of costs (US$76; SD=70). *P. vivax* had the highest mean total household cost at US$199 (SD=174), followed by *P. knowlesi* and *P. falciparum* at US$119 (SD=81 and SD=83, respectively), and *P. malariae* (US$99; SD=42). Most patients (80%) experienced direct health costs above 10% of monthly income, with 58 (38%) patients experiencing health spending over 25% of monthly income, consistent with catastrophic health expenditure.

**Data availability statement:** All relevant data used for analysis of the manuscript and supporting information are available via attached online repository (https://github.com/PatrickAbraham1/Sabah-Knowlesi-COI). This includes the minimum data set and coding for all means, standard deviations, and values used to create figures.

**Funding:** This work was supported by the National Health and Medical Research Council, Australia (Grant Numbers 1037304 and 1045156 to NMA; 1042072 fellowship to NMA; investigator grants 2017436 to MJG and 2025362 to AD; the Australian Centre of Research Excellence in Malaria Elimination grant 1134989 to NMA and MJG and grant 2024622 to MJG and AD); The MONKEYBAR project (UK Environmental & Social Ecology of Human Infectious Diseases Initiative, Grant #G1100796 to CD); the National Institutes of Health, USA (R01AI160457-01 to TW and GSR); Malaysian Ministry of Health (Grant Number BP00500/117/1002 to GSR), the Australian Centre for International Agricultural Research, and Indo-Pacific Centre for Health Security, DFAT, Australian Government (LS-2019-116 to MJG and AD). The funders had no role in the study design, data collection and analysis, decision to publish, or preparation of the manuscript.

**Competing interests:** The authors have declared that no competing interests exist.

## Conclusions/significance

Despite Malaysia's free health-system care for malaria, patients and families face other related medical, travel, and indirect costs. Household out-of-pocket costs were driven by productivity losses; primarily attributed to infections in working-aged males in rural agricultural-based occupations. Costs for *P. vivax* were higher than those of *P. knowlesi* and *P. falciparum*. This may be attributable to a younger age profile and the longer treatment required to clear the liver-stage parasites of *P. vivax*.

## Author summary

Malaria patients experience financial barriers when seeking care, despite free access to universal care in Malaysia. In some areas of Southeast Asia, multiple species of malaria can be present within the same setting, including recently emerged knowlesi malaria, which is spread from monkeys to humans via mosquitos. The economic burden of illness due to multiple species of malaria has not previously been estimated in the same setting. We collected data on the cost of illness to households in Sabah, Malaysia, to estimate their related total economic burden during the elimination phase of human-only malaria. Medical costs and time off work and usual activities were substantial in patients with the four species of malaria diagnosed during the time of this study. This research highlights the financial burden which households face when seeking care for malaria in Malaysia, despite the free treatment provided by the government.

## Introduction

Malaria is a vector-borne disease caused by infection with parasitic protozoa of the *Plasmodium* genus, which continues to exert a high health burden and cost in endemic settings [1,2]. Six *Plasmodium* species commonly cause malaria in humans: *P. falciparum, P. vivax, P. malariae, P. ovale, P. wallikeri, P. curtisi* and *P. knowlesi* [3]. Since *P. falciparum* and *P. vivax* have historically caused the most significant health burden in humans, elimination efforts have primarily focused on these species [4]. While *P. malariae* cases commonly co-exist with *P. falciparum* and *P. vivax*, these cases tend to have very low parasitemia and mild disease severity [5]. Malaysia is now nearing elimination of non-zoonotic malaria with no indigenous cases of *P. falciparum* and *P. vivax* reported in the country for the last 5 years [1].

Malaria carries a significant financial burden to both health systems and the individuals, families, and communities experiencing the disease [2]. Malaysia has free universal access to malaria screening, testing, and treatment through public health facilities [6,7]. However, households still incur out-of-pocket costs for travel, medicines, and healthcare visits, which, to our knowledge, have not been previously estimated. High out-of-pocket payments required for healthcare services can potentially cause adverse consequences for patients and their families, often disproportionally affecting low-income households [8]. Similarly to the elimination efforts, most of the economic literature has focused on the costs of *P. falciparum* and *P. vivax*, typically portraying the costs of a single malaria species [2,9]. One previous study that compared *P. vivax* and *P. falciparum* household costs in Indonesia found these to be similar [10]. No previous studies have estimated the cost of *P. knowlesi* or *P. malariae* infections, nor compared these costs to other species of malaria found within the same area.

The emergence of zoonotic *P. knowlesi* transmission since 2004 has been of particular concern as it threatens progress towards World Health Organization malaria elimination goals in Southeast Asia. *P. knowlesi* infections occur in areas inhabited by natural macaque hosts where it is transmitted to humans via *Anopheles* Leucosphyrus Group mosquito vectors [11]. *P. knowlesi* is now the major cause of malaria in Malaysia, with 3000 to 4000 malaria case notifications per year since 2017 [12–14]. *P. knowlesi* infections carry a high risk of severe disease, with 6-9% of symptomatic cases presenting to health facilities in East Malaysia classified as severe malaria, comparable with previous risk for *P. falciparum* (5-11%) in sympatric areas [12,15–17].

*P. knowlesi* is most prevalent in males aged between 20 and 40 years in Malaysia, who are the most commonly reported demographic in formal employment [15,18,19]. Males within this age group are more susceptible to acquisition of zoonotic malaria due to outdoor agricultural work, such as farming, forestry, palm oil and rubber plantations, in rural areas near forest edges [20,21]. More than half of *P. knowlesi* cases nationally have been attributed to agriculture and plantation-based work activities, with many of the remaining cases linked to timber logging and forestry work [22]. Due to the lack of effectiveness of traditional malaria public health control methods for zoonotic transmission primarily at the forest-edge [20,21], the need for inter-sectoral approaches has been recommended for *P. knowlesi* [22].

The increase in the incidence of *P. knowlesi* [22] is primarily in the East Malaysia states of Sabah and Sarawak on the island of Borneo [23]. Sabah, the nation's third most populous state, consists predominantly of rural agricultural areas with tropical rainforest and large areas of mountainous terrain, which combined with large-scale deforestation primarily for oil-palm plantations and rural population growth have created an environment suitable for zoonotic malaria transmission [19,24,25]. In 2017 Sabah accounted for 45% of all reported malaria cases in Malaysia, with an almost five times greater annual case-incidence than the national average [26].

This study used a household perspective to measure and value out-of-pocket costs and productivity losses for a single malaria episode as part of a survey collected at health facilities in Sabah, Malaysia between 2013 and 2016, a period during which malaria was caused by four *Plasmodium* species, including zoonotic *P. knowlesi*.

## Methods

### Ethics

The study was approved by the Medical Research Ethics Committee of the Ministry of Health, Malaysia (NMRR-12-537-12568) and the Human Research Ethics Committee of Menzies School of Health Research, Australia (HREC-2012-1814). All adult participants provided written informed consent, with parental/guardian written informed consent gained for participants less than 18 years of age.

### Participants and study design

The cost surveys were collected alongside a larger epidemiological study conducted in Sabah, Malaysia, between February 2013 and September 2016 [20]. In line with national guidelines in Malaysia, all microscopically confirmed malaria cases required mandatory hospitalization for clinical management, including antimalarial drug treatment until at least two sequential microscopy slides are negative for malaria. Participants were included based upon an initial microscopic diagnosis of malaria, with final *Plasmodium* species determined by the laboratory PCR result. The patient population had an even spread across socio-economic quartiles, consistent across rural districts in Sabah [20]. A subset of patients presented initially to primary

clinics prior to referral to the appropriate district or tertiary level hospital for admission. Patients were recorded as having severe malaria as classified by WHO research criteria [27]. These criteria included one or more of the following: impaired consciousness, acidosis, hypoglycemia, severe malarial anemia, acute kidney injury, jaundice, respiratory distress, significant bleeding, shock, or hyperparasitemia (defined as a parasite count >100,000/μL) [28].

A retrospective, cross-sectional cost survey was conducted across three district hospitals: Kudat, Kota Marudu, and Pitas, and one tertiary referral hospital in Kota Kinabalu. A pretested, structured, closed-ended questionnaire [10] was directly administered to all patients at clinics by trained research nurses to malaria patients or caregivers at around 28 days after hospital admission to estimate the costs of a single malaria episode [20]. No payments were made to participants.

## Data collection and valuation of productivity losses

Household out-of-pocket costs were reported in Malaysian Ringgits. The out-of-pocket costs incurred by a household from a single malaria case was estimated using an ingredients-based approach, by individually questioning expenditure on travel, hospitalization, clinic visits, medication and lost productivity. Total household costs were presented as aggregate costs of direct, indirect, and other costs for a single episode of malaria. Other costs included any additional expenses that patients incurred during their illness. Patients were not asked about the source of these costs.

Direct costs include all out-of-pocket expenses a patient incurred for diagnosis, hospital stays, clinic visits for follow-up monitoring, medical treatments, inpatient food or drink, over-the-counter medication, and transport while seeking medical care. Direct costs were used to determine catastrophic health expenditure, determined as direct, out-of-pocket expenditures while seeking treatment exceeding 10% and 25% of total monthly household income [29]. Two alternate sources of monthly income (Malaysian Household Income Survey and Household Inequities Survey [30,31]) were also used as separate thresholds to calculate catastrophic health expenditure and to estimate the sensitivity of those results.

Indirect costs for patients were expressed as lost productivity, reported in the survey as "days unable to work (or decreased work) or go to school" due to the malaria episode. Indirect costs were measured using a human capital approach [32]. Reduced work hours due to illness was the sum of survey respondents absent time from work prior to presenting to hospital, the number of days hospitalized, and the number of recovery days after hospital discharge when patients were unable to work. For children who were below the minimum age of employment (16 years), only the productivity losses for caregivers were included. The value of a day of lost wages was self-reported in the survey, and for adults who reported lost time but did not disclose income (34% of adults), the mean wage of the recorded income from all others was used.

Indirect costs for caregivers were the number of days when additional care was needed for the patient and/or if the patient required someone else to care for their dependents. Caregiver costs were estimated using a proxy good approach [33]. We valued caregiver days lost as equivalent to 50% of the average reported lost daily wage for patients. Since 66% (63/96) of patients in this dataset who reported being looked after by someone were also unable to look after their own dependents, the reduced wage estimate has been used to avoid double counting. This conservative approach minimizes the potential overestimation of caregiving as it is likely the same caregiver could be used for the patient and their dependents.

## Data analysis

Patient data analysis was performed using STATA statical software, version 16 (StataCorp LP, College Station, TX, USA). Costs were inflated using gross domestic product (GDP) deflators

[34] before conversion to 2023 US dollars (US$), equating to an approximate conversion of 4.4 MYR to 1 USD$ [35]. The mean and standard deviations (SD) were calculated, and uncertainty for household costs was handled by performing a univariate sensitivity analysis of key parameters. Mann-Whitney tests were used to identify statistically significant differences in costs between groups with two outcomes, such as sex, and severe and non-severe malaria. A Kruskal-Wallis test was used to identify differences in household costs between individuals with malaria grouped by the four *Plasmodium* species [36]. Additionally, a generalized linear model (GLM) was used to model the marginal effect on total cost of severe malaria, sex, anemia status, human-only malaria status and age (while holding all other variables constant) using a gamma-distributed dependent variable with a log link, as appropriate for this data [37]. Likely confounders were based on prior knowledge, and plausibility of variables which may have impacted costs.

For the evaluation of lost productivity, five scenario analyses were performed. In Scenario 1, productivity loss costs were limited to those patients who self-reported lost wage earnings. In Scenario 2, local wage estimates derived from a government-validated source (Malaysian Household Income Survey [30]) was used for all adults. For Scenario 3 we applied the local wage estimate to all participants, including children. For Scenarios 4 and 5, the lost wage was taken from a modelled regression analysis of the household income survey controlling for rurality and ethnicity (Malaysian Household inequities survey [31]) and applied to adults only and then to all participants, respectively.

## Results

### Socio-demographic characteristics of study participants

Data were collected for 152 participants presenting with a confirmed episode of malaria. The mean length of time from enrolment to performing the interview was 31 days. Characteristics of the study participants are presented in Table 1. The mean age was 32 years old (SD = 18), and 75% (114/152) were male. The majority of patients were diagnosed with *P. knowlesi* infections (71%, 108/152). Seven (5%) patients met the WHO research criteria for severe malaria; all were due to *P. knowlesi* infection. Most patients (73%, 110/151) had at least mild anemia (WHO age/sex criteria [38]) during their malaria episode, though the anemia status was not known for one patient. No patients died during this study.

The mean monthly income for all working-age adults who reported income (N=80) was US$158.78 (SD = 126.24).

### Time losses due to malaria illness

Overall, the average number of days participants were unable to complete their usual activities due to illness was 9.2 (SD = 7.0) (Table 2). *P. vivax* (11.0 days) and *P. falciparum* (10.3 days) had a higher number of days affected by illness compared to *P. knowlesi* (8.7 days) and *P. malariae* (8.5 days). The average number of days lost to illness increased to 13.9 (SD = 15.6) for severe *P. knowlesi* cases, comparatively higher than non-severe *P. knowlesi* cases at 8.3 days (SD = 6.5, p = 0.141) and all non-zoonotic malaria (*P. falciparum*, *P. vivax* and *P. malariae*) cases at 10.4 days (SD = 5.80, p = 0.477).

Patients reported an average length of stay in hospital of 4.1 days (SD=1.4; range 1-10). The length of hospitalization for severe *P. knowlesi* cases (mean = 5.6 days, SD = 1.4) was higher than for non-severe malaria cases due to any species (mean = 3.9, SD = 1.4, p=0.004). Three patients (2%) were admitted within hospital to a high dependency clinical unit for a mean of 3.3 days; however, of these only a single patient was subsequently categorized as severe malaria using research criteria.

**Table 1. Demographic and disease characteristics of study population (N = 152).**

|  | n | % |
| --- | --- | --- |
| **Sex (male)** | 114 | 75% |
| **Age, mean (SD)** | 31.8 (18.1) | - |
| **Working age (16-65 years)** | 110 | 72% |
| **Age distribution** |  |  |
| <5 years | 4 | 3% |
| 5-15 years | 30 | 20% |
| 16-35 years | 64 | 42% |
| 36-50 years | 26 | 17% |
| 50-65 years | 20 | 13% |
| >65 years | 8 | 5% |
| **Employment status** |  |  |
| Employed | 77 | 51% |
| Unemployed | 8 | 5% |
| Housewife | 7 | 5% |
| Unknown | 18 | 12% |
| Student | 38 | 25% |
| Child younger than school age (5 years) | 4 | 3% |
| **Species of malaria** |  |  |
| *P. knowlesi* | 108 | 71% |
| *P. vivax* | 22 | 14% |
| *P. falciparum* | 16 | 11% |
| *P. malariae* | 6 | 4% |
| Severe malaria | 7 | 5% |
| **Anemia status (n=151)\*** |  |  |
| Anemic | 110 | 73% |
| Non-anemic | 41 | 27% |
| **District of residence** |  |  |
| Pitas | 53 | 37% |
| Kota Marudu | 41 | 28% |
| Kudat | 32 | 23% |
| Penampang | 8 | 6% |
| Kota Kinabalu | 4 | 3% |
| Papar | 3 | 2% |
| Ranau | 3 | 2% |
| Other | 1 | 0.7% |
| **Site of hospital admission** |  |  |
| Pitas | 49 | 32% |
| Kota Marudu | 46 | 30% |
| Kudat | 32 | 21% |
| Kota Kinabalu | 25 | 17% |

SD = standard deviation.

\* Hemoglobin data needed to calculate anemia were missing from one patient.

Eighty-eight patients (58%) reported being unable to fulfill their role as primary caregiver for their household for a mean of 3.9 days during their illness, indicating additional care was needed for their dependents. In addition, 97 patients (64%) reported that another household member supported them for a mean of 4.8 days after leaving hospital. The mean amount

**Table 2. Mean (standard deviation) number of days away from usual activity per patient and their household due to an episode of malaria (N=152). All severe cases in the study were due to *Plasmodium knowlesi*.**

| | P. falciparum (n=16) | P. vivax (n=22) | P. malariae (n=6) | Non-severe P. knowlesi (n=101) | Severe P. knowlesi (n=7) | Total |
|---|---|---|---|---|---|---|
| **Time away from usual activity** | | | | | | |
| Days away from usual activity pre-hospital[1] | 2.3 (3.1) | 2.9 (3.6) | 1.67 (3.20) | 1.6 (2.2) | 3.7 (7.7) | 2.0 (3.0) |
| Duration of hospital stay | 4.4 (1.0) | 4.9 (1.3) | 3.66 (0.82) | 3.9 (1.4) | 5.6 (1.4) | 4.1 (1.4) |
| Days away from usual activity post-hospital[1] | 3.6 (5.9) | 3.1 (3.6) | 3.17 (3.19) | 2.8 (5.9) | 4.6 (7.5) | 3.0 (5.6) |
| ***Total days patient unable to do usual activities*** | **10.3 (6.7)** | **11.0 (5.5)** | **8.5 (4.50)** | **8.3 (6.5)** | **13.9 (15.6)** | **9.2 (7.0)** |
| **Time required for caregiving** | | | | | | |
| Caregiving received | 2.9 (3.0) | 5.3 (10.5) | 2.3 (2.0) | 2.5 (2.2) | 5.6 (7.2) | 3.1 (4.8) |
| Alternative caregiving required[2] | 1.8 (1.9) | 3.1 (3.7) | 2.1 (1.8) | 2.3 (2.0) | 4.1 (3.1) | 2.5 (2.4) |
| ***Total days of caregiving*** | **4.8 (3.7)** | **8.4 (11.5)** | **4.4 (2.7)** | **4.9 (3.2)** | **9.7 (9.1)** | **5.6 (5.7)** |
| **Total days impacted[3]** | **15.1 (8.7)** | **19.3 (12.5)** | **12.9 (5.6)** | **13.2 (7.2)** | **23.6 (16.9)** | **14.8 (9.2)** |

[1]Days away from usual activity was expressed in the survey as "days unable to work (or reduced work) or unable to attend school".

[2]Alternative caregiving was expressed in the survey as "did anyone else have to look after your dependents?"

[3]Total days impacted is the sum of total days unable to work and total days caregiving.

of time spent by patients travelling was 1.2 days to seek treatment before plus 1.6 days after leaving the hospital to attend follow-up clinic visits (Table A in S1 Text). Travel days were not included in the total days impacted by the household, as this would duplicate the time spent away from usual activities. Distance from health facility was not collected within the survey.

## Household costs of malaria

Direct household costs equated to US$53.89 (SD=66.20) per person (41% of the total cost). The majority of direct costs were related to the hospital admission, although it is unclear whether these costs included direct admission costs or cost associated food and other supplies needed during an admission. Indirect household costs were the most significant contributor to the total cost burden (58%), with a mean total of US$75.84 (SD=69.67) for each household. When using self-reported wages (with mean wage imputed for adults who reported time off usual activities but not their income), the mean patient time cost was estimated to be US$53.60 (SD=58.77). The total caregiver time cost (including patients being looked after themselves and when alternative caregiving was needed to look after the patient's dependents) was estimated to be US$22.24 (SD=28.40) per malaria episode. Only 3% (4/152) of patients reported having "other costs" which were unspecified in the survey. These were <1% of the total costs, all were *P. knowlesi* cases, and three of these were reported by female patients. The mean total household cost per malaria episode was US$130.53 (SD=102.24) with a median of $96.11 (IQR=$70.73, $153.01) (Table 3).

## Costs of malaria by population subgroup

Male participants reported a slightly greater average cost of illness with US$134.26 compared to US$119.35 for females; however, this difference was statistically non-significant (Table 4). Cost differences between sexes were primarily attributed to reported wage differences, with 65% (25/38) of females not reporting their wage losses. This group predominantly consisted of females identifying as houseworkers with no formal paid employment (13/38) or students under 16 years (10/38). Despite higher overall cost for men, females had 45% higher direct costs of US$70.26 compared to US$48.44 for men (p=0.018), attributed to higher hospital and clinic costs (Table C in S1 Text). Costs were also compared between working-age males (16-65

**Table 3. Total mean and median household costs for a single malaria episode in 2023 United States dollars (n=152). Results in Malaysian Ringgits are presented in Table B in S1 Text.**

| | Mean | Standard deviation | Percent of total cost | Median | Interquartile range |
|---|---|---|---|---|---|
| Hospital admission | 26.46 | 41.11 | 20% | 12.63 | 5.09 – 35.68 |
| Clinic visit | 14.19 | 24.41 | 11% | 7.65 | 5.05 – 12.74 |
| Medicine | 2.65 | 11.41 | 2% | 0.00 | 0.00 – 0.92 |
| Travel to clinic and hospital | 10.60 | 17.81 | 8% | 5.10 | 0.00 – 12.74 |
| *Total direct costs* | **53.89** | **66.20** | **41%** | **32.23** | **20.44 – 55.69** |
| Patient time | 53.60 | 58.77 | 41% | 36.76 | 14.15 – 66.16 |
| Caregiver time | 22.24 | 28.40 | 17% | 17.80 | 7.35 – 26.63 |
| *Total indirect costs* | **75.84** | **69.67** | **58%** | **55.55** | **30.35 – 95.56** |
| *Total other costs*[1] | **0.80** | **5.92** | **0.6%** | **0.00** | **0.00 – 0.00** |
| **Total costs** | **130.53** | **102.24** | **100%** | **96.11** | **70.73 – 153.01** |

[1]Other costs were unspecified in the questionnaire; no details about the nature of these costs were provided by the participants.

**Table 4. Total mean household costs and standard deviations (SD) for a malaria episode by sex, species, and malaria severity in 2023 United States dollars (n=152).**

| | N | Direct costs | | Indirect costs | | Other costs | | Total | | Total costs p-value |
|---|---|---|---|---|---|---|---|---|---|---|
| | | Mean | SD | Mean | SD | Mean | SD | Mean | SD | |
| **Sex** | | | | | | | | | | 0.330 |
| Male | 114 | 48.44 | 56.01 | 85.38 | 76.18 | 0.45 | 4.76 | 134.26 | 103.18 | |
| Female | 38 | 70.26 | 89.07 | 47.22 | 30.98 | 1.86 | 8.51 | 119.35 | 99.86 | |
| **Malaria species** | | | | | | | | | | 0.404 |
| *P. knowlesi* | 108 | 45.38 | 44.54 | 73.43 | 65.50 | 1.12 | 7.01 | 119.94 | 81.39 | |
| *P. vivax* | 22 | 109.33 | 127.99 | 89.72 | 88.20 | 0.00 | – | 199.05 | 173.87 | |
| *P. falciparum* | 16 | 42.044 | 37.01 | 77.65 | 79.65 | 0.00 | – | 119.69 | 82.68 | |
| *P. malariae* | 6 | 35.45 | 32.07 | 63.52 | 43.44 | 0.00 | – | 98.98 | 41.85 | |
| **Malaria severity** | | | | | | | | | | 0.006 |
| Non-severe | 145 | 52.70 | 65.00 | 71.73 | 64.61 | 0.84 | 6.06 | 125.27 | 99.13 | |
| Severe | 7 | 78.65 | 90.21 | 160.98 | 114.33 | 0.00 | – | 239.64 | 112.53 | |
| **Hospital** | | | | | | | | | | 0.001 |
| Kota Marudu | 46 | 34.36 | 26.03 | 60.20 | 58.06 | 0.00 | – | 94.57 | 62.14 | |
| Kudat | 32 | 73.24 | 48.84 | 71.22 | 67.41 | 1.81 | 9.03 | 146.28 | 82.33 | |
| Pitas | 49 | 64.76 | 97.26 | 75.59 | 72.70 | 1.30 | 7.47 | 141.64 | 138.02 | |
| Kota Kinabalu | 25 | 43.78 | 52.17 | 111.00 | 77.39 | 0.00 | – | 154.78 | 89.40 | |

years old, which comprised 54% of the total participants) and the remainder of those enrolled. The total cost for working-age males was higher than the remainder of the participants, with a mean of US$146.34 (SD=106.90) versus US$112.01 (SD=93.88), respectively (p=0.006; Table D in S1 Text). While direct costs were similar between working-age males and the rest of the population, working-age males reported higher indirect costs at US$95.66, compared to US$52.62 for others (p<0.001).

The seven severe malaria cases (all *P. knowlesi*) had 91% higher total costs than non-severe cases (US$239.64 versus US$125.27 respectively, p=0.006) (Table 4), predominantly due to substantially higher indirect costs (US$89.25 mean difference) for severe malaria. *P. vivax* patients presented with the highest total costs at $199.05 (SD=173.87; Table 4). In contrast to other

malaria species, most of the *P. vivax* costs were direct costs (63%) where hospital and clinic out-of-pocket costs were double the direct costs of malaria due to other *Plasmodium* species (Table E in S1 Text). *P. falciparum* and *P. knowlesi* had similar mean costs of US$119.69 (SD=82.68) and US$119.94 (SD=81.39), respectively. *P. malariae* episodes had the lowest total cost at $98.98 (SD=41.85). Additional Mann-Whitney pairwise comparisons were performed between each *Plasmodium* species for direct, indirect, and total costs; no results were statistically significant. A further breakdown by species and cost category is presented in Table E of S1 Text.

## Multivariable analysis of malaria costs

The GLM tested severe malaria, sex, anemia status, human-only malaria status and age against total costs. The GLM showed that when holding all other variables constant, severe malaria has an incremental marginal cost of US$58.39 (95% CI= -$19.86, $136.63) when compared to non-severe malaria (Table 5). Having a human-only type of malaria lead to an increase in marginal costs of $50.23 (95% CI= $14.86, $85.60), while a one year increase in age lead to an increase in marginal costs of $1.49, when controlling for other factors. This result was consistent with a *P. knowlesi* only GLM where age was the only significant predictor with marginal cost of $1.11 (95% confidence interval $0.18, $2.03; p=0.019; Table F of S1 Text). Outputs of the GLM are presented in Table G of S1 Text.

## Scenario analysis

The wage used to estimate productivity losses greatly impacted the total cost per malaria episode. Compared to the base case estimate of US$130.53 per person, the cost per episode decreased to US$114.30 when wage loss estimates were only applied to adults who reported their wages for Scenario 1 (Table 6). When using Sabah-specific estimates from the Malaysia Household Income survey (US$223.57 per month) applied to only adults in Scenario 2 [30], the total costs increased by 19% to US$158.85. When this wage was applied to the time loss of all patients in Scenario 3, the total cost increased to US$178.61, an increase of 37% compared to the base case. Finally, when the household inequities survey for Sabah was utilized (US$389.21 per month) [31], the total cost increased by 78% (US$236.03) when applied to all adults in Scenario 4 and 104% (US$270.41) when applied to all patients in Scenario 5. Females had higher total costs than males for Scenarios 3 and 5 where wage estimates were applied evenly to the entire population (Fig 1).

## Catastrophic health expenditure

When using the mean monthly self-reported income value of US$148, direct health expenditure for a single malaria episode accounted for 27% of monthly income. Eighty percent

**Table 5. Marginal costs, standard errors, and 95% confidence intervals of factors associated with variability with total household costs from the generalized linear model that included all patients (N=151). All costs are in 2023 United States dollars.**

|  | Marginal cost from base case (dy/dx) | Standard error | P-value | 95% confidence interval |
|---|---|---|---|---|
| Sex (Male) | 18.73 | 18.70 | 0.317 | -17.93, 55.38 |
| Age | 1.49 | 0.53 | 0.005 | 0.45, 2.52 |
| Severe malaria | 58.39 | 39.92 | 0.144 | -19.86, 136.63 |
| Human-only malaria* | 50.23 | 18.05 | 0.005 | 14.86, 85.60 |
| Anemia | 15.61 | 18.76 | 0.405 | -21.15, 52.38 |

*Includes falciparum, vivax, and malariae malaria

**Table 6. Scenario analysis of mean and median indirect and total household costs for a single malaria episode using varied wages to estimate productivity losses in 2023 United States dollars (n=152).**

| | Mean | SD | % of total cost | Median | Interquartile range |
|---|---|---|---|---|---|
| **Base case: Reported incomes with mean income applied to all adults who did not report income** | | | | | |
| Total Indirect Costs | 75.84 | 69.67 | 56% | 55.55 | 30.35 – 95.56 |
| Total Costs | 130.53 | 102.24 | | 96.11 | 70.73 – 153.01 |
| **Scenario 1: Only reported incomes considered** | | | | | |
| Total Indirect Costs | 59.61 | 71.12 | 46% | 23.40 | 18.97 – 69.01 |
| Total Costs | 114.30 | 106.28 | | 78.69 | 49.13 – 137.94 |
| **Scenario 2: MHIS wage only applied to adults** | | | | | |
| Total Indirect Costs | 104.16 | 81.69 | 66% | 92.35 | 51.75 – 134.55 |
| Total Costs | 158.85 | 115.28 | | 132.58 | 86.60 – 189.07 |
| **Scenario 3: MHIS wage applied to everyone** | | | | | |
| Total Indirect Costs | 123.91 | 79.42 | 69% | 103.50 | 79.41 – 144.91 |
| Total Costs | 178.61 | 112.23 | | 147.03 | 111.88 – 201.24 |
| **Scenario 4: HIS wage applied to adults** | | | | | |
| Total Indirect Costs | 181.33 | 142.22 | 77% | 160.77 | 90.09 – 234.25 |
| Total Costs | 236.03 | 168.82 | | 203.44 | 125.03 – 285.70 |
| **Scenario 5: HIS wage applied to everyone** | | | | | |
| Total Indirect Costs | 215.72 | 138.26 | 80% | 180.19 | 138.25 – 252.27 |
| Total Costs | 270.41 | 163.77 | | 220.20 | 180.03 – 315.34 |

MHIS: Malaysia Household Income Survey [30], $223 per month

HIS: Household Inequities Survey [31], $389 per month

(122/152) of patients reported direct health costs above 10% of monthly income, indicating catastrophic health expenditure. When using a higher defined threshold for catastrophic health expenditure of 25% of monthly income, the direct costs of a malaria episode reported by 38% (57/152) of patients in this survey remained at levels consistent with catastrophic health expenditure.

When using the Malaysia household income survey data as a higher baseline mean monthly income value of US$223 [30], 62% (94/152) of patients reported health expenditure above 10% of monthly income. This percentage dropped to 24% (36/152) when applying the higher 25% monthly income defined threshold. If considering the mean monthly income from the household inequities survey of US$389 [31], then 39% (59/152) and 12% (18/152) of patients experience catastrophic health expenditure at 10% and 25% monthly income thresholds, respectively.

## Discussion

Despite the provision of free health-facility-related costs of malaria care by the Ministry of Health, Malaysia [39], patients in Sabah faced substantial costs when accessing care with a mean of $131 for each malaria episode. Malaysia has successfully controlled human-only malaria in recent years, with only small numbers of imported cases of *P. vivax* and *P. falciparum* now reported [1]. However, transmission of four major *Plasmodium* species were diagnosed during the period this study was conducted, with the majority being due to local zoonotic *P. knowlesi* infections. To our knowledge, this is the first study to present and compare costs for more than two species of human-only malaria and the first to compare costs between zoonotic and non-zoonotic malarias.

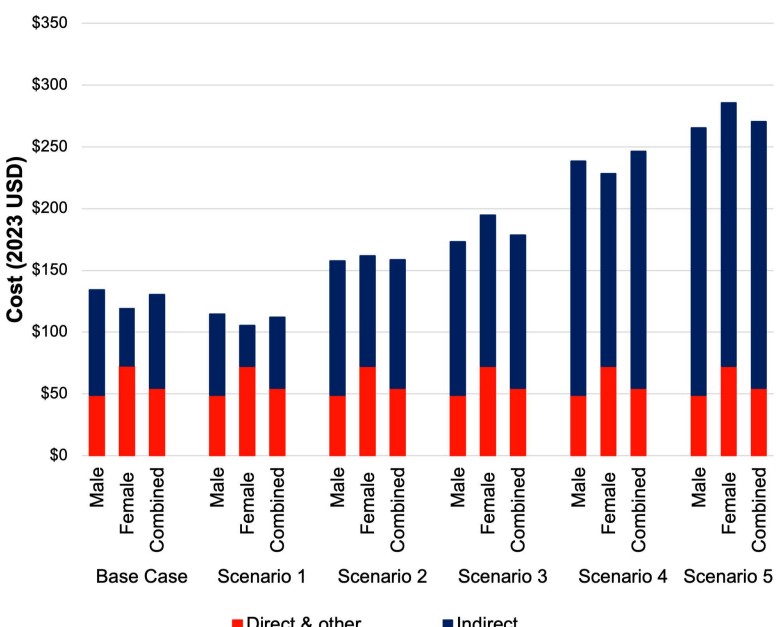

**Fig 1. Scenario analysis of methods for estimating the mean indirect costs and associated impact on the total cost per malaria episode by sex in 2023 United States dollars (n =152).** The base case uses reported incomes with mean income applied to all adults who did not report income. Scenario 1 uses only reported incomes to value productivity losses. Scenario 2 applies the Malaysian Household Income Survey wage [30] to adults only. Scenario 3 applies the Malaysian Household Income Survey wage [30] to all patients including children. Scenario 4 applies the Household Inequities Survey wage [31] to adults only. Scenario 5 applies the Household Inequities Survey wage [31] to all patients including children.

One of the major findings was that patients with *P. vivax* infections demonstrated a trend towards higher mean costs of US$199, mainly due to direct medical costs for higher-intensity treatment monitoring [40]. The higher cost associated with *P. vivax* infections compared to other *Plasmodium* species has a number of potential contributing factors related both to the patient and their care. *P. vivax* symptomatic infections predominantly occurred in a younger population demographic, with 46% of vivax malaria participants under 15 years, meaning relatively lower productivity costs for the patients and higher costs for their caregiving. The younger age profile of vivax malaria patients may explain the higher direct out-of-pocket medical costs associated with the disease as parents and caregivers of these patients will likely travel, sleep, and eat in the ward alongside the child. Another factor which may contribute to higher costs is caregivers may be more likely to return their child for follow-up clinic services, and *P. vivax* requires a longer treatment duration of 14 days for the primaquine course, in contrast to other species that do not have dormant liver parasites to treat [6]. However, the survey did not collect data on the number of clinic visits for each patient. To our knowledge, no previous studies have shown the cost of *P. vivax* infections to be higher than *P. falciparum* infections, however this should be interpreted cautiously due to the low case numbers of *P. falciparum*.

As expected, severe malaria was associated with higher costs (US$240) than non-severe disease (US$125). This is consistent with the available literature for non-severe and severe *P. falciparum* infections in African contexts [41,42], despite severe cases being exclusively due to *P. knowlesi* infection in this study. Females within this study had higher direct costs than men at $70 as compared to $48, respectively. While the source of this difference was not captured by

our survey, the study team suggested that this may be explained by higher spending by females on hygiene products while in hospital. When using the mean reported monthly income from the study, 28% of patients experienced catastrophic health expenditure at a threshold of 25% of monthly income. This is consistent with endemic settings in Africa, where catastrophic health expenditure due to malaria have ranged from 18-22%; however, these were in vastly different health and economic contexts [43–45]. To our knowledge, no catastrophic health expenditure estimates for malaria have been previously estimated in Asian settings.

Compared with infections due to non-zoonotic *Plasmodium* species, patients diagnosed with *P. knowlesi* were older and more likely to be male, reflecting the current understanding of transmission occurring among adult males engaged in agricultural activities [20,46,47]. Overall, working-aged males (18-65 years), showed lower total direct medical costs than those of non-working age. This may be attributed to employers subsidizing out-of-pocket costs, such as large-scale plantation owners providing free transport.

Factors found on multivariable analysis to remain independently associated with higher total costs were age and having malaria from human-only *Plasmodium* species (compared to *P. knowlesi* infection). Both male sex and severe malaria did not remain as independent predictors when controlling for these other features. Understanding the distance travelled by participants to access care and being able to control for this may have been useful in this model. However, these data were not collected within the survey. Without further data, it is challenging to speculate on other factors associated with higher total costs per malaria episode. However, when controlling for malaria severity, malaria from human-only *Plasmodium* species (predominantly influenced by higher *P. vivax* related costs) was associated with higher total costs (Table 5), which may be reflective of the longer duration of illness with *P. vivax* and *P. falciparum* compared with non-severe *P. knowlesi* (Table 2).

While government health services were provided at no charge for malaria patients, out-of-pocket costs for households still have the potential to impact those experiencing socio-economic disadvantage disproportionally. Whilst this study did not directly capture socioeconomic status, understanding occupational status and district of residence may provide insight into subpopulations experiencing disadvantage. Irrespective of the method used to estimate wages and the health expenditure threshold chosen, a large proportion of malaria patients exceeded the catastrophic health expenditure threshold for mean direct out-of-pocket costs. It is conceivable that for many lower-income households, dealing with the impact of this type of health expenditure may necessitate either deferring initial or subsequent care, particularly considering the self-reported income from participants in the study was far less than the estimated average incomes from the region. Alternatively, households may deplete savings, take funding for schooling, sell cash crops or animals, or incur other forms of significant debt, collectively contributing to a possible long-term detrimental household debt-cycle [1]. Many of these factors may be further exacerbated if occurring in conjunction with a limited household savings buffer, including a lack of cash flow for those reliant on subsistence farming, for which selling assets such as land or changing agricultural practices to prioritize growing of higher risk cash crops may therefore further jeopardize future household earning potential or stability [29,48]. Households with lower socio-economic status have been found to be at a higher risk of having malaria [10], indicating the inequitable burden these households bear.

The scenario analyses showed significant variation in total costs when considering alternative methods to value time losses. The base case scenario imputed the mean self-reported income for adults with no income-reported data. If only evaluating those with self-reported data on productivity losses (Scenario 1), the lower subsequent malaria cost estimates are likely to underestimate lost productivity for subsistence farmers and the informal workforce [49,50]. Since self-reported income may not be the most appropriate method, we also used

alternative wage estimates to value these time losses, applying them primarily to all adults and then as a comparator to the whole population (including children). Using separate analyses based on the large robust Malaysian Household Income Survey [30], and health inequities reporting data [31] to provide alternative estimates of the average monthly wage to all adults, the total cost of illness increased by 19% and 78% respectively. Valuation of productivity losses is a highly discussed methodological issue [51], particularly considering income differences between male and female respondents [52]. By using standardized wage estimates, time losses from all patients are considered equal, and the total cost reflects an equity-driven estimate.

Notably, in Scenarios 3 and 5 (when the two alternative wage estimates were uniformly applied to all patients), the total costs were higher for females than males. Valuing children's time using a cost equal to adult wages, while not representative of formal economic output, gives greater insight into the economic cost of time losses due to malaria in children, given that malaria episodes have been shown to be detrimental to children's education [53–55]. In these scenarios, productivity losses account for 70% and 80% of total costs, respectively, compared to the base case where indirect costs accounted for 58% of total costs. The method by which time losses are valued can greatly alter the interpretation of the total household costs of malaria. Given the often overlooked value of the informal economy [49], scenarios which value all patients time losses equitably emphasize the true opportunity cost faced by patients and families during a malaria episode.

This study has several limitations. The self-reported costing data was collected 30 days after study enrolment; hence, potentially neglecting the long-term chronic health costs, especially for the low numbers of *P. vivax* patients who are at risk of relapse infections [56]. Costs of other longer-term complications were not included due to feasibility limitations [15]; however, these costs may conceivably have a large effect on actual household costs. Case numbers were also low for *P. vivax*, *P. falciparum* and *P. malariae* infections. Additionally, recall bias and administering the survey as part of a research setting may have resulted in under- or over-estimation of costs and resources spent by a household related to a single malaria episode [57]. Females may also be underrepresented within the sample due to gender-related socio-cultural norms that influence female's treatment-seeking behavior [58]. Additionally, no questions were directly asked about total household income, thus if households have multiple income sources, they have been excluded from the catastrophic health expenditure calculations. No questions within the survey directly asked patients about household financial stability, which limited our ability to explore this aspect. This study aimed to estimate costs to the households; therefore, costs to the healthcare provider were not included. The inclusion of healthcare provider costs would further increase the total cost per malaria episode, and future studies should quantify costs from a societal perspective to provide a more comprehensive evaluation of the economic burden of malaria. Lastly, the data in this study is dated, with no indigenous cases of *P. falciparum* and *P. vivax* present in the area, which limits the applicability of these findings, however there are still imported cases.

## Conclusion

Given Malaysia's goal of malaria elimination, understanding the magnitude of household out-of-pocket costs and productivity losses is essential to inform policy, guide novel interventions and enable future cost-effectiveness analyses. Patients face a substantial financial burden for each episode of malaria, even when health-facility based clinical management is otherwise provided free of charge. These costs can disproportionately impact low socio-economic households that rely on subsistence farming and agricultural-related income, with potentially catastrophic economic impacts if ignored. To our knowledge, this is the first study exploring household costs of malaria in Malaysia and the first to quantify the economic burden

of *P. malariae* and *P. knowlesi* episodes on households, which are poorly understood. This information is crucial to the East Malaysian states of Sabah and neighboring Sarawak, which collectively report the majority of *P. knowlesi* cases nationally. *P. knowlesi* transmission is now demonstrated to occur in other locations in Southeast Asian locations [11]; without adequate control measures, the medical costs and indirect productivity losses will continue to place a substantial economic burden on patients and their families.

## Supporting information

**S1 Text. Supplementary information.**
(DOCX)

## Acknowledgments

The authors thank the study participants, the IDSKKS malaria research team, and the Sabah State Health Department for assisting in the conduct of this study. We thank the Director General of Health Malaysia for permission to publish this manuscript.

## Author contributions

**Conceptualization:** Timothy William, Nicholas M Anstey, Matthew J Grigg, Angela Devine.

**Data curation:** Campbell McMullin, Roddy Teo, Chris Drakeley.

**Formal analysis:** Patrick Abraham, Jenarun Jelip, Roddy Teo, Chris Drakeley, Abdul Marsudi Manah.

**Investigation:** Jenarun Jelip.

**Methodology:** Nicholas M Anstey, Matthew J Grigg, Angela Devine.

**Project administration:** Jenarun Jelip, Roddy Teo, Chris Drakeley, Abdul Marsudi Manah.

**Resources:** Timothy William, Giri S Rajahram, Nicholas M Anstey.

**Supervision:** Timothy William, Giri S Rajahram, Nicholas M Anstey, Matthew J Grigg, Angela Devine.

**Writing – original draft:** Patrick Abraham, Campbell McMullin, Matthew J Grigg, Angela Devine.

**Writing – review & editing:** Patrick Abraham, Chris Drakeley, Nicholas M Anstey, Matthew J Grigg, Angela Devine.

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
