## [Decision Letter · Decision Letter 0]

24 Nov 2024

PNTD-D-24-00626The economic burden of zoonotic Plasmodium knowlesi malaria on households in Sabah, Malaysia compared to malaria from human-only Plasmodium speciesPLOS Neglected Tropical Diseases Dear Dr. Abraham, Thank you for submitting your manuscript to PLOS Neglected Tropical Diseases. After careful consideration, we feel that it has merit but does not fully meet PLOS Neglected Tropical Diseases's publication criteria as it currently stands. Therefore, we invite you to submit a revised version of the manuscript that addresses the points raised during the review process. Please submit your revised manuscript within 60 days Jan 23 2025 11:59PM. If you will need more time than this to complete your revisions, please reply to this message or contact the journal office at plosntds@plos.org. Please include the following items when submitting your revised manuscript: * A rebuttal letter that responds to each point raised by the editor and reviewer(s). You should upload this letter as a separate file labeled 'Response to Reviewers'. This file does not need to include responses to any formatting updates and technical items listed in the 'Journal Requirements' section below. * A marked-up copy of your manuscript that highlights changes made to the original version. You should upload this as a separate file labeled 'Revised Manuscript with Track Changes'. * An unmarked version of your revised paper without tracked changes. You should upload this as a separate file labeled 'Manuscript'. If you would like to make changes to your financial disclosure, competing interests statement, or data availability statement, please make these updates within the submission form at the time of resubmission. Guidelines for resubmitting your figure files are available below the reviewer comments at the end of this letter. We look forward to receiving your revised manuscript. Kind regards,Amanda RossAcademic EditorPLOS Neglected Tropical Diseases Abhay SatoskarSection EditorPLOS Neglected Tropical Diseases

Shaden Kamhawi

co-Editor-in-Chief

Paul Brindley

co-Editor-in-Chief

**Additional Editor Comments :** Please consider all of the reviewers' comments below. In addition, I have some comments on the analysis for consideration. The text p15 on subgroups has some statistical errors. L232 “statistically insignificant” – do you mean non-significant? ‘Insignificant’ would change the meaning and imply that it is not important, non-significant would convey that p>0.05. L244 “p=0.000” – this p-value cannot be zero, although it can be given as 0.00 in the software output due to rounding. It is usually written for example ‘p<0.001.’ L231: It is misleading to state that “there are differences but they are not significant”. There is simply no evidence of a difference and the difference that you observe is likely to have arisen by chance.  Please be careful of wording for statistical concepts throughout the manuscript.

The sample size is very small for some of the malaria species (n=6 for *P. malariae*, n=16 for* P. falciparum, *n=22 for *P. vivax*, n=108 for *P. knowlesi*). This could be more prominently acknowledged and borne in mind while interpreting the results.

Did you include all patients presenting between February 2013 to September 2016? If not, how were they selected?

It is not obvious to see how you can disentangle all the factors with this sample size. As one reviewer mentions, the male and female differences observed could be due to residual confounding. I suspect you are making more of this than is justified by the data.

The patients came from three hospitals – the costs may vary between hospitals (and this may cause a lack of statistical independence between observations), however hospital does not seem to have been taken into account in the analysis.

There are many p-values in this manuscript. Doing many tests can lead to the problem of multiple comparisons. For one hypothesis test, if there is no difference (the null hypothesis is true), there is still a 5% chance of a false positive. If you have do many tests, then the risk of at least one false positive increases. Did you consider this?

Please give a measure of uncertainty (eg SD, 95% CI etc) when giving point estimates (eg a mean, a measure of the strength of an association).

In a formal document strictly speaking, ‘knowlesi’ would be ‘P. knowlesi’.

‘multivariate analysis’ – I think you mean ‘multivariable analysis’. Multivariate has multiple outcomes whereas multivariable has multiple explanatory variables.

**Journal Requirements:**

1) Please upload all main figures as separate Figure files in .tif or .eps format. For more information about how to convert and format your figure files please see our guidelines: 

2) We have noticed that you have uploaded Supporting Information files, but you have not included a list of legends. Please add a full list of legends for your Supporting Information files after the references list.

3) We note that your Data Availability Statement is currently as follows: "All relevant data are within the manuscript and its Supporting Information files". Please confirm at this time whether or not your submission contains all raw data required to replicate the results of your study. Authors must share the “minimal data set” for their submission. PLOS defines the minimal data set to consist of the data required to replicate all study findings reported in the article, as well as related metadata and methods (https://journals.plos.org/plosone/s/data-availability#loc-minimal-data-set-definition).

4) Please amend your detailed Financial Disclosure statement. This is published with the article. It must therefore be completed in full sentences and contain the exact wording you wish to be published.

2) State what role the funders took in the study. If the funders had no role in your study, please state: "The funders had no role in study design, data collection and analysis, decision to publish, or preparation of the manuscript.".

5) Please ensure that the funders and grant numbers match between the Financial Disclosure field and the Funding Information tab in your submission form. Note that the funders must be provided in the same order in both places as well. Currently, this grant number [1134989] provided by " the Australian Centre of Research Excellence in Malaria Elimination" is missing from the Funding Information tab.

Please indicate by return email the full and correct funding information for your study and confirm the order in which funding contributions should appear. Please be sure to indicate whether the funders played any role in the study design, data collection and analysis, decision to publish, or preparation of the manuscript.

**Reviewers' Comments:**Reviewer's Responses to Questions

**Key Review Criteria Required for Acceptance?**

**Methods**

-Are the objectives of the study clearly articulated with a clear testable hypothesis stated?

-Is the study design appropriate to address the stated objectives?

-Is the population clearly described and appropriate for the hypothesis being tested?

-Is the sample size sufficient to ensure adequate power to address the hypothesis being tested?

-Were correct statistical analysis used to support conclusions?

-Are there concerns about ethical or regulatory requirements being met?

Reviewer #1: Methods

-Are the objectives of the study clearly articulated with a clear testable hypothesis stated? Yes

-Is the study design appropriate to address the stated objectives? Yes

-Is the population clearly described and appropriate for the hypothesis being tested? Yes

-Is the sample size sufficient to ensure adequate power to address the hypothesis being tested? Yes

-Were correct statistical analysis used to support conclusions? Yes

-Are there concerns about ethical or regulatory requirements being met? Yes

Reviewer #2: The study presents a clear objective to use a household perspective in measuring and valuing out-of-pocket (OOP) expenses and productivity losses for episodes of malaria, disaggregated by malaria type. The study design is sound; however, I suggest adding more details on data collection methods (e.g., whether it involved clinic visits or phone interviews) and clarifying the patient selection criteria (e.g., was it every person presenting at the hospitals during the study period or a specific subset?).

Comment 1: Please provide more information on how patients were contacted—whether through phone or in-person clinic visits—and clarify if participants received compensation for their time.

Comments 2: Additionally, describe the questions used to collect cost data, especially regarding admissions or clinic costs. Did you ask a single question (e.g., "What did you pay in total?") or inquire about each cost component individually (e.g., “What did you pay for transport?” “What did you pay for food?”)? In the results (Line 212), you mention, "The majority of direct costs were related to hospital admission," which is unclear about whether these included only admission fees or also associated costs like food and supplies, suggesting a single-question approach.

Comment 3: Line 124: Please specify further what the two sources of monthly income are.

Comment 4: Line 128: Did you use the human capital approach? My understanding is that a consistent value for time lost (e.g., $X per day) is applied to all individuals, regardless of occupation. You note in the results (Line 23) that "Cost differences between sexes were primarily attributed to reported wage differences, with 65% (25/38) of females not reporting their wage losses.” This discrepancy is what the human capital approach aims to mitigate by assigning a standard value to time. Additionally, you mention (Line 127) including missed school hours; how did you value these, as they are not income-related?

Comment 5: Line 162: Regarding the GLM, what was the rationale for choosing the specified variables? Did you conduct any model diagnostics to assess fit?

Reviewer #3: 4. I found the “study design and participants” section difficult to follow. Considering separate paragraphs for “study design” and “participants.”

**Results**

-Does the analysis presented match the analysis plan?

-Are the results clearly and completely presented?

-Are the figures (Tables, Images) of sufficient quality for clarity?

Reviewer #1: -Does the analysis presented match the analysis plan? Yes

-Are the results clearly and completely presented? Yes

-Are the figures (Tables, Images) of sufficient quality for clarity? Yes

Reviewer #2: The analysis presented aligns well with the stated analysis plan, and the results are both clearly and comprehensively presented. The figures and tables provided are of high quality, making the findings accessible and easy to interpret.

Comment 6: The results section is very clear overall. To enhance clarity and alignment, I suggest structuring the presentation of costs similarly to the methods section. Presenting the direct out-of-pocket (OOP) expenditures first, followed by productivity losses, could provide a smoother transition into a combined "Household Cost of Malaria" summary.

Comment 7: In Table 6, adding a per-month salary for each scenario would provide additional context for readers, allowing a clearer understanding of productivity loss within the scope of household economics.

Comment 8: The scenario analysis is an excellent addition to the paper and effectively highlights the assumptions driving productivity loss estimates. However, the paragraph presenting the scenario results could be made clearer. Consider reorganizing this information into a table to enhance readability—this could be incorporated into an existing table (e.g., Table 6) or presented in a new one.

Reviewer #3: No concerns

**Conclusions**

-Are the conclusions supported by the data presented?

-Are the limitations of analysis clearly described?

-Do the authors discuss how these data can be helpful to advance our understanding of the topic under study?

-Is public health relevance addressed?

Reviewer #1: -Are the conclusions supported by the data presented? Yes

-Are the limitations of analysis clearly described? Yes

-Do the authors discuss how these data can be helpful to advance our understanding of the topic under study? Somewhat

-Is public health relevance addressed? Somewhat

Reviewer #2: The conclusions are well-supported by the data presented, and the discussion section effectively summarizes the main findings, addressing key questions raised throughout the paper. The authors also highlight the study’s public health relevance, emphasizing the importance of understanding malaria’s economic impact on households.

Comment 9: Regarding the higher costs for P. vivax, did you collect data on the number of clinic visits for each family? This information could substantiate your claim in Line 350 that P. vivax cases involved more frequent clinic visits. Additionally, in explaining the cost difference, you mention that P. vivax predominantly affects younger patients. However, you also state that older patients tend to incur higher costs; it would be helpful to reconcile this point in the discussion.

Comment 10: In Line 340, consider comparing your findings on catastrophic health expenditure to other studies. Discuss whether your findings align with, exceed, or fall below similar estimates in the literature, providing context to enhance the relevance of this result.

Comment 11: Line 398-400 appears to have missing words. Please review and clarify this sentence to ensure the intended meaning is conveyed.

Comment 12: Line 431: When mentioning that the study setting may have influenced cost estimates, please elaborate. Specify how the setting could have impacted the findings and in which direction (e.g., increased or decreased costs).

Comment 13: Line 442: Consider rephrasing this sentence to emphasize the utility of cost-of-illness data in informing cost-effectiveness analyses. Being explicit about this can clarify its relevance for policymakers and researchers.

Reviewer #3: 5. Because the species of malaria infection correlates with patient characteristics (e.g. people with P. knowlesi are more likely to be men), I’m left wondering whether it is the type of malaria infection or the characteristics of the patients driving these differences in cost. Are you able to disentangle that at all?

**Editorial and Data Presentation Modifications?**

Reviewer #1: Reader may benefit from having more visual presentation of data/results - at the moment the results are primarily detailed in tables.

Reviewer #2: I’ve provided detailed comments for the methods, results, and discussion/conclusions sections, including editorial suggestions and recommendations for data presentation. In addition, I have some suggestions below for the introduction, which is overall well-written and provides a strong foundation for the study.

Comment 14: On Line 51-52, it may be clearer to report the number of malaria cases directly instead of using the term “public health notifications,” which could be confusing for readers unfamiliar with the phrasing.

Comment 15: Line 54: Please include the corresponding percentage for P. falciparum cases. Additionally, the sentence structure suggests that all patients presenting at a health facility have severe disease, which is not consistent with the information provided later. Clarifying this here would avoid any misinterpretation.

Comment 16: Line 78-80: Consider rephrasing to: “However, households still incur out-of-pocket costs for travel, medicines, and healthcare visits, which, to our knowledge, have not been previously estimated.” This phrasing improves clarity and flow.

Comment 17: Line 47: You mention that there have been no indigenous cases of P. falciparum or P. vivax in five years, but several cases were observed in this study. If this discrepancy is due to the study period (2013–2016), please clarify in the discussion to address potential reader confusion.

Reviewer #3: Nothing to add here

**Summary and General Comments**

Reviewer #1: General comments: A thorough and thoughtful costing exercise. I appreciated the transparency of the methods and clearly the authors spent time investigating the quirks of their data (e.g. the anecdote to explain the cost differences between males/females). I think the methods and results are all strong and comprehensive. Though the conclusions could be strengthened to drive home the importance of the findings to reinforce statements like "Our research also highlights theopportunity cost of not developing novel public health interventions to control P. knowlesi transmission." This would be strengthened by a simple multiplication of the patient costs per case and an estimate of the number of knowlesi cases per year in Sabah. Presumably these costs will be used (or would be useful) in cost-effectiveness studies in Sabah/Malaysia of interventions from a societal perspective and more fully capture the impact of these interventions - would be good to state this.

Main comments:

i) The data were collected between 2013 and 2016, this is quite a while ago. I think there needs to be some acknowledgement of this and brief discussion of the generalisability to the situation today in Malaysia.

ii) Would be useful to understand more about the economic situation of Sabah, how does it sit comparatively in terms of wealth to other parts of Malaysia?

iii) The authors stated that the analysis is from a household perspective yet household earnings were not measured, only the patient's. The authors acknowledged the catastrophic health expenditure calculation is performed only using the proportion of the individuals earnings. However, other members of the households costs were included (e.g. caring responsibilities etc). So the costs are accrued from the entire household yet divided by the individuals earnings? If I have understood that correctly then this would likely overestimate the cases of catastrophic health expenditure. Is it possible to use alternative estimates of average household, rather than individuals, incomes in Sabah as the denominator? At the very least this should be explored as a scenario analysis. Or if just the individual's earnings are being used as the threshold, then perhaps only the individuals costs should be used as the numerator (though the perspective would then be the patient's perspective rather than households).

iv) I appreciate data on socioeconomic status were not collected, however it would be interesting if the authors can infer something about SES from the data they do have to help describe the patients who are presenting. E.g. from the reported earnings data vs Sabah/Malaysian average wages.

v) This study will probably be published in 2025, yet the reported costs are in 2022 USD. Could the authors inflate these costs to 2024 values?

vi) The introduction states malaria testing and treatment is free under the UHC program in Malaysia yet there are direct medical costs included in the results - this appears to be a contradiction to me and seems an important point to clarify, are these copayments under the insurance scheme? Are these patients for whatever reason ineligible from insurance coverage (e.g. due to nationality - was that captured?). The authors state "it is unclear whether these costs included direct admission costs or cost associated food (sic) and other supplies needed during an admission" - My sense is it seems a bit high for just food costs alone. It is also entirely possible that hospitals levy fees despite policies in place but it would be good to clarify the case here if possible. The combined possible direct medical costs make up $45.76 (33%) of the total and may be avoidable by policy/enforcement of policy whereas to some extent productivity losses can only be reduced by reducing the incidence of malaria

Minor comments:

i) Page 19, line 306: "When using the mean monthly self-reported income value of US$168, direct health expenditure for a single malaria episode accounted for 27% of monthly income." The $168 is different to the value reported earlier in the paper ($156) - is this because this is the inflated value? Needs clarification.

ii) Page 22, line 399: "indicating the inequitable these households bear." Sentence needs adjusting.

iii) Page 24, line 436: "they have been missed in catastrophic health expenditure calculation" Sentence needs adjusting.

Reviewer #2: The manuscript, “The economic burden of zoonotic Plasmodium knowlesi malaria on households in Sabah, Malaysia compared to malaria from human-only Plasmodium species,” provides a novel and valuable contribution to understanding the financial impact of malaria at the household level, particularly by differentiating costs by malaria parasite type. This study fills an important gap in the literature by highlighting the economic burden of malaria in Malaysia and has potential implications for both public health and economic policy.

The paper is clearly written, logically structured, and effectively guides the reader through its key findings and implications. The finding that P. vivax incurs higher costs than other malaria types may seem unexpected, but the authors do an excellent job of presenting plausible explanations for this in the discussion, addressing factors such as age distribution and frequency of clinic visits among patients with P. vivax. Their discussion supports the robustness of the study’s conclusions.

The methodological execution is rigorous, and the inclusion of both direct and indirect household costs is commendable. However, two areas could benefit from further clarification. First, the scenario analyses, while methodologically sound, could be made more interpretable for the reader by enhancing the presentation in the results section, possibly through more detailed description or an additional table summarizing the scenarios. Second, more explanation on the variable selection in the modeling process would strengthen the paper. Specifically, clarifying whether the model was theory-based, derived from backward selection etc. would provide insight into the modeling rationale and improve transparency. Also the authors do not mention any regression model diagnostics are used to assessed the validity and reliability of a model's results.

These minor revisions would enhance the clarity and interpretability of an already strong manuscript. Congratulations to the authors for their thorough work and significant contributions to understanding malaria’s economic impact on households. I look forward to seeing these revisions in the final version.

Reviewer #3: 1. In my view, the most interesting/important finding was that most patients are incurring catastrophic health expenditures regardless (if I’m understanding it correctly) of the species of malaria infection and despite the country’s policy of covering malaria treatment expenses. I don’t know the Malaysian context well, but I suspect that if malaria elimination is the goal, there is already motivation to address P. knowlesi. The fact that P. knowlesi costs are not very different from other malaria costs may just reinforce the status quo. It could be a more impactful paper if it has a health financing message rather than a malaria control message.

2. If you opt to keep the current framing, the Background section needs to make a stronger case as to why comparing the costs of different species of malaria is important, and better justify the need for the subgroup analysis. Including hypotheses would be helpful. What cost differences did you expect there to be, and why?

Some of this content could be relocated from the Discussion, which is quite long. There is also some contextualization in the Results (lines 256-7), which should be moved elsewhere.

3. This is a small concern, but under “Ethics,” it would be better to list all of the IRB’s than to say the IRB’s included XYZ.

PLOS authors have the option to publish the peer review history of their article (what does this mean?). If published, this will include your full peer review and any attached files.

Reviewer #1: No

Reviewer #2: No

Reviewer #3: **Yes: **Janna M. Wisniewski

---

## [Decision Letter · Decision Letter 1]

29 Jan 2025

PNTD-D-24-00626R1Higher economic burden of Plasmodium vivax compared to P. falciparum and zoonotic P. knowlesi on households in Sabah, MalaysiaPLOS Neglected Tropical Diseases Dear Dr. Abraham, Thank you for submitting your manuscript to PLOS Neglected Tropical Diseases. After careful consideration, we feel that it has merit but does not fully meet PLOS Neglected Tropical Diseases's publication criteria as it currently stands. Therefore, we invite you to submit a revised version of the manuscript that addresses the points raised during the review process. Please submit your revised manuscript within 30 days Feb 28 2025 11:59PM. If you will need more time than this to complete your revisions, please reply to this message or contact the journal office at plosntds@plos.org. Please include the following items when submitting your revised manuscript: * A rebuttal letter that responds to each point raised by the editor and reviewer(s). You should upload this letter as a separate file labeled 'Response to Reviewers'. This file does not need to include responses to any formatting updates and technical items listed in the 'Journal Requirements' section below. * A marked-up copy of your manuscript that highlights changes made to the original version. You should upload this as a separate file labeled 'Revised Manuscript with Track Changes'. * An unmarked version of your revised paper without tracked changes. You should upload this as a separate file labeled 'Manuscript'. If you would like to make changes to your financial disclosure, competing interests statement, or data availability statement, please make these updates within the submission form at the time of resubmission. Guidelines for resubmitting your figure files are available below the reviewer comments at the end of this letter. We look forward to receiving your revised manuscript. Kind regards, Amanda RossAcademic EditorPLOS Neglected Tropical Diseases Abhay SatoskarSection EditorPLOS Neglected Tropical Diseases

Shaden Kamhawi

co-Editor-in-Chief

Paul Brindley

co-Editor-in-Chief

**Additional Editor Comments:** Please respond to the small comment from Reviewer 2

Some of the responses to comments have introduced some confusion:

- Reviewer 3 had commented that the paper may have a stronger message if the paper has a health financing message rather than a malaria control message. The authors responded by changing the title to compare P vivax to the other malarias (rather than the original version comparing P knowlesi to the other malarias) on the basis that P vivax tends to have the higher costs. This does not address the comment directly and has the unfortunate side-effect that the title is now inconsistent and jars with the introduction, results and discussion which focus on the comparison of P knowlesi to P vivax and P falciparum.

Please change either the title or the text so that they are consistent, and consider whether the message should include a health financing angle.

- Forward selection of variables for the GLM

I could find one value of AIC in the legend for table S7. However it is not clear how forward selection was carried out - was there a cut-off for the AIC change used for selecting variables? There are some non-significant variables included in the table and in the GLM. Forward selection is not necessary or recommended, simply considering likely confounders from prior knowledge would be fine. However if forward selection is mentioned, it ought to be clear what the criterion for inclusion was.

- A previous comment on the sample size was been responded to but rather indirectly. Please check that it is clearly stated that the sample size in this study was small for P vivax, P malariae and P falciparum.

- L273 - Lower bound of the CI is $-19 but in the table it is -119. The very wide CI in the table for the GLM suggest that the GLM is squeezing the small sample size rather hard or perhaps that the gamma distribution does not fit so well. It would be useful to check the statistics. Some of the CI bounds are not obviously plausible.

- Standard deviations are included in the abstract for the overall costs however they are also needed for the species-specific costs.

- L254 p=<0.001 typo (it should be p<0.001)

- The reasons for the tendency for higher costs of P vivax are not consistent - in the discussion the different reasons are nicely explained, however in the abstract the message is different and one reason only is implicated. **Journal Requirements:**

Please ensure that the funders and grant numbers match between the Financial Disclosure field and the Funding Information tab in your submission form. Note that the funders must be provided in the same order in both places as well.

**Reviewers' comments:** Reviewer's Responses to Questions

**Key Review Criteria Required for Acceptance?**

**Methods**

-Are the objectives of the study clearly articulated with a clear testable hypothesis stated?

-Is the study design appropriate to address the stated objectives?

-Is the population clearly described and appropriate for the hypothesis being tested?

-Is the sample size sufficient to ensure adequate power to address the hypothesis being tested?

-Were correct statistical analysis used to support conclusions?

-Are there concerns about ethical or regulatory requirements being met?

Reviewer #1: (No Response)

Reviewer #2: (No Response)

**Results**

-Does the analysis presented match the analysis plan?

-Are the results clearly and completely presented?

-Are the figures (Tables, Images) of sufficient quality for clarity?

Reviewer #1: (No Response)

Reviewer #2: (No Response)

**Conclusions**

-Are the conclusions supported by the data presented?

-Are the limitations of analysis clearly described?

-Do the authors discuss how these data can be helpful to advance our understanding of the topic under study?

-Is public health relevance addressed?

Reviewer #1: (No Response)

Reviewer #2: (No Response)

**Editorial and Data Presentation Modifications?**

Reviewer #1: (No Response)

Reviewer #2: The authors have adequately answered all of my questions and comments from the first round of review. One small suggestion - please report if participants received compensation for their time.

**Summary and General Comments**

Reviewer #1: The authors have addressed the comments where appropriate and provided detailed responses with good justifications where they have declined to make changes. I am satisfied with the updated manuscript.

Reviewer #2: (No Response)

PLOS authors have the option to publish the peer review history of their article (what does this mean?). If published, this will include your full peer review and any attached files.

Reviewer #1: No

Reviewer #2: **Yes: **Katherine Snyman

---

## [Editor Report · Decision Letter 2]

25 Feb 2025

Dear Mr Abraham,

We are pleased to inform you that your manuscript 'Household costs associated with zoonotic Plasmodium knowlesi, P. falciparum, P. vivax and P. malariae infections in Sabah, Malaysia' has been provisionally accepted for publication in PLOS Neglected Tropical Diseases.

Best regards,

Amanda Ross

Academic Editor

Abhay Satoskar

Section Editor

Shaden Kamhawi

co-Editor-in-Chief

Paul Brindley

co-Editor-in-Chief

---

## [Editor Report · Acceptance letter]

Dear Mr Abraham,

We are delighted to inform you that your manuscript, "Household costs associated with zoonotic Plasmodium knowlesi, P. falciparum, P. vivax and P. malariae infections in Sabah, Malaysia," has been formally accepted for publication in PLOS Neglected Tropical Diseases.

Best regards,

Shaden Kamhawi

co-Editor-in-Chief

Paul Brindley

co-Editor-in-Chief
